# Online informal learning community for interpreter training amid COVID-19: A pilot evaluation

Da Yan◉, Qiongqiong Fan*◉

School of Foreign Languages, Xinyang Agriculture and Forestry University, Xinyang, China

◉ These authors contributed equally to this work.
* 2012280003@xyafu.edu.cn

## Abstract

Sudden shifts towards online education since the outbreak of Covid-19 propelled the unprepared changes in teaching and learning over the world. The impact of transferring Interpreter training from face-to-face instruction and practices to a fully online environment was viewed differently. Issues such as relatively inferior engagement in learning and dissatisfied performance in competence building were highlighted and compounded by the concern of academic burnout and learning stress caused by the abiding pandemic. To curb the unsatisfactory situation, alternative learning methods and innovative pedagogical approaches were advocated. The present study was a pioneering effort to integrate informal learning into remote interpreter training by developing and implementing an online informal learning communities for undergraduate interpreting trainees in a Chinese university. The researcher recruited 36 students (n = 36) from the institution as participants in the 1.5-year piloting project. The findings of the research revealed the impact of informal learning in supplementing formal education by engaging involved students. Student-centered learning supported by collaborative and experiential activities in an informal environment was well-received for its ability to galvanize student's engagement and academic achievements. The perceptions from participants revealed preference and expectation from students for expanded roles of trainers in interpreter training.

**Data Availability Statement:** All relevant data are within the paper and its Supporting Information files.

**Funding:** The authors received no specific funding for this work.

## 1. Introduction

The outbreak of Covid-19 has changed education across the globe dramatically. Face-to-face instructions were prohibited in many institutions due to lockdowns of campus and shutdown of public facilities, consequently, teaching and learning were forced to shift to Emergency Remote Teaching (ERT) [1]. The sudden shift resulted in issues in the readiness of lecturers and institutions for maintaining educational quality amid the pandemic [2]. Large scale online teaching called for higher level of digital literacy and technology competences from teachers as well as proactivity and engagement from students [3]. Relatively weak internet infrastructure in some areas also restricted the quality and availability of online learning [4].

**Competing interests:** The authors have declared that no competing interests exist.

As online teaching and learning proceeded, underlying issues of the educational system such as regional differences in allocation of educational resources and communication infrastructures were amplified during Covid-19 [5, 6]. Additionally, new issues such as the emotional and psychological wellbeing of learners and instructors during the pandemic became a major concern [7]. It was argued that "burnout, depression, and social media addiction" were directly connected to Covid-19 related psychological issues [8]. Extensive use of digital devices and prolonged duration of exposure to online learning environment would deteriorate the stress and burnout encountered by learners [9].

Despite all the difficulties faced by educators suffering from threatened efficacy, the pandemic also brought hope and opportunity for innovations in education and post-pandemic resilience [10]. For example, the combination of informal learning and online education was implemented in pioneering efforts in several documented cases to offset the negative impact of the pandemic [11–13].

## 1.1 Interpreter training in the pandemic

In pre-Covid-19 eras, remote Learning or Distance Interpreter training (DIT) were practiced with fruitful outcomes. In a 13 week remote training program, students' learning achievements supported the claim that distance mode of interpreter training could bring students to a similar level as face-to-face mode [14] In a medical interpreter training program, Virtual Interpreter training and Learning (VITL) was successfully implemented to provide widely accessible and economically affordable training for learners over the world [15].

Nonetheless, since the outbreak of Covid-19 and consequent changes in educational policies and the emergence of new social norms, the teaching and learning of interpreting via distance teaching altered correspondingly. In accordance with the limited number of existing research on Translation and Interpreting (T&I) teaching and learning amid the pandemic, the efficacy and strength of DIT were diversified and occasionally doubted [16]. In a national survey on the status of T&I training during the pandemic in Slovakia, three major factors in remote learning were identified: procedural, technical and psychosocial factors [17]. Due to the nature of T&I education, the transfer of student-centered drills and practices in face-to-face education to online teaching were challenging. Students' learning engagement and motivation were the main concern of T&I education amid Covid-19, and many instructors were challenged by inability to organize effective learning community among students in the online learning context [18]. In a survey on an undergraduate Interpreting and Translation education in Korea in 2020, respondents reported that psychological and emotional distress experienced from learning would hinder the feedback of students' progress and stop students from getting adapted to the new learning environment [19]. However, the pandemic also ushered in positive stimulus for the advancement in education technology and the evolvement of T&I education. For example, flipped classroom teaching model was well-claimed for courses on translation theories if the teaching was well organized and structured [20].

In China, where officially accredited T&I education were provided as part of the bachelor's and master's degree programs, experiences were gained from attempts to conduct ERT for trainee students from hundreds of universities. In a comparative study on the perceptions towards DIT in several universities across China, most respondents agreed that the pandemic and resulting changes in the training of interpreters promoted the development of online and blending approaches for interpreter training [21]. Contrarily, in a survey for students' preferences for face-to-face mode versus distance learning mode of interpreter training, opinions and attitudes of students suggested that remote learning were still underappreciated due to limitations in the delivery approaches, learning environment and effectiveness [22].

Currently, there existed a knowledge gap in understanding the impact of Covid-19 on the teaching and learning of T&I education throughout the world. As the impact of the pandemic continued, efforts should be made in systematically understanding the impact of alternative learning method adopted during Covid-19 in T&I education and best practices recommended for interpreting programs for post-pandemic resilience.

## 1.2 Informal learning

As students' engagement and mental health were prioritized as major concerns in education amid Covid-19 [23], education policy maker and curriculum designers began experimenting new learning approaches and innovative pedagogical methods to sustain the motivation and involvement of students during the pandemic [24, 25]. Informal learning was an answer to the growing knowledge density and need for innovation learning methods when the position of conventional formal learning was challenged for its inefficiency in adopting to the new situations [26]. Informal learning was frequently practiced in workspaces where higher level of professionalism was valued and it could be used to facilitate the innovation of employee's work behavior [27].

The value of informal learning was emphasized for its ability to support and reinforce formal learning while contributing to enhance outcomes in cognitive, practical and affective domains of student-directed learning [28]. For its flexibility, universities educators' perception towards the strength of informal learning grew with the deepened understanding of its nature and power in assisting formal education [29, 30]. The combination of information technologies and informal learning were believed to be able to maximize learning outcomes. In a research on the use of WhatsApp as a medium of informal learning in the daily lives of teenagers, the results confirmed a boosting role of web-based informal learning in developing participants' skills and learning abilities [31]. Fusing informal learning with other learning strategies was also becoming a trend in innovative attempts to create new learning environments. Chinese L2 learners in an informal learning environment with gamification experienced enhancement in learning achievement and motivation from the rather joyful environment [32].

Nevertheless, there was a dearth of research or practices of informal learning implemented in interpreter training. According to existing literatures as well as observations of interpreter training in many institutions, informal learning was rarely practiced. Hence, the understanding of the impact of informal learning in reinforcing formal interpreter training was generally unknown. However, the impact of informal learning in enhancing students' engagement, psychological satisfaction, learning autonomy, and academic achievements posed a possible route to answer the challenges faced by online interpreter training during Covid-19 [33, 34].

## 1.3 The study

Against the backdrop of the knowledge gap in the theoretical, practical and conceptual understanding of implementing online informal learning in interpreter training amid Covid-19, the present study set out to investigate the impact of online informal learning in enhancing students' learning engagement and academic achievements by piloting a project of informal learning since the campus lockdowns in early 2020. The objective of the study was to examine and consolidate the understanding of the development and implementation of online informal learning to assist formal interpreter training during and after the pandemic.

## 2. Methods

This section described the context for the study, the informal leaning community and methodologies adopted in the research.

## 2.1 Context

The study was conducted at Xinyang Agriculture and Forestry University (XYAFU), an undergraduate university in China. The Bachelor of Arts in Translation and Interpreting (BTI) program at the institution was approved and established since September 2018. The BTI program was accredited by Ministry of Education, People's Republic of China. By March of 2022, there were approximately 450 students receiving T&I training as BTI candidates.

The teaching and learning of the BTI program at XYAFU were impacted by the outbreak of Covid-19 since late 2019. From the outbreak of the pandemic to present, XYAFU was forced to adopt ERT fully or partially for four consecutive semesters. Being a relatively new BTI program, the teaching and learning at XYAFU was vulnerable to public health emergencies due to the lack of backup plans or alternative pedagogical methods. The sudden shift towards online teaching resulted in a number of unexpected outcomes, for both lecturers and students.

## 2.2 Informal learning community

The informal learning community was a loosely organized student-directed learning model for BTI candidates at XYAFU. The pilot project to test its impact to promote ERT teaching amid Covid-19 kicked off since September 2020, after a rather dissatisfied semester. Lecturers for core translation and interpreting courses (n = 7) were voluntarily recruited to construct and guide the online learning community. Two male lecturers were recruited while five other participating lecturers were female; one participant was of bachelor's degree in arts, six with a master's degree in Translation and Linguistics. Table 1 demonstrates the demographical information of lecturer participants. Up to March 2022, students and lecturers participated in four sessions of informal learning, each per semester.

In the informal learning community, the starting point was the continuum of formal interpreter training (either as face-to-face instruction or remote teaching during the pandemic). The informal learning community adhered to a "one core three pillars" design. The core of the informal learning community was formative assessment. Student-directed learning, collaborative learning and experience sharing were the three pillars within the system. Fig 1 showed the conceptual model of the informal learning community.

Formative assessment served as the core of the informal learning community. Assessment was not to provide summative and judgmental appraisal of students' performance in interpreting tasks, but to facilitate the learning with immediate feedbacks [35]. Students took frequent peer-assessment and recurrent self-assessment for tasks and projects they have completed. Larger scale assessments and lecturer-guided assessments were less frequently practiced but in necessity. Results and findings from all assessments were synthesized by students as documentaries or portfolios of the learning activities.

**Table 1. Demographic information of participating lecturers.**

| Name | Gender | Age | Education Background | Expertise | Experiences (years) |
|------|--------|-----|----------------------|-----------|---------------------|
| Lec. 1 | M | 31 | MA in Translation | English/Chinese Interpreting | 7 |
| Lec. 2 | F | 28 | MA in Translation | Interpreting for Special Purpose | 3 |
| Lec. 3 | F | 29 | MA in Translation | Consecutive Interpreting | 2 |
| Lec. 4 | F | 45 | BA in Linguistics | English/Chinese Interpreting | 12 |
| Lec. 5 | M | 29 | MA in Translation | Simultaneous Interpreting | 4 |
| Lec. 6 | F | 28 | MA in Translation | Interpreting for Business | 3 |
| Lec. 7 | F | 27 | MA in Translation | Interpreting Technologies | 3 |

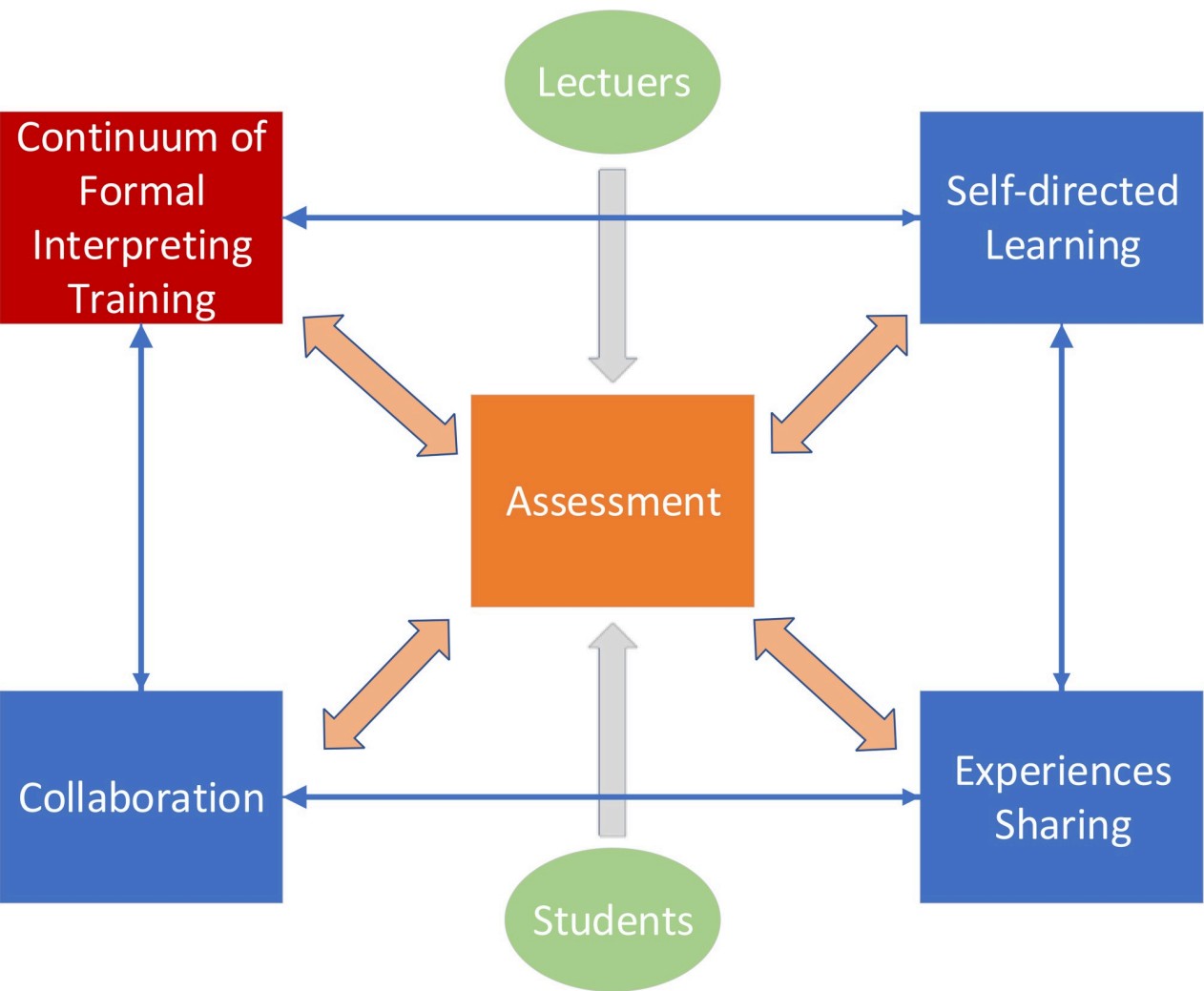

**Fig 1. Conceptual model of informal learning community.**

The learning process was self-regulated as all critical decisions made during the learning process were left for the students. Students were allowed to start, modify and terminate their learning voluntarily. The open-ended engagement was hypothesized by the researcher to be effective in raising students' interest and motivation during the learning. With the help of web applications and virtual conference applications, the collaboration between students in learning played a pivotal. At indefinite intervals, students and lecturers were invited to join in experience sharing brainstorms to summarize and digest the gains and pitfalls in learning.

Without a set schedule or time frame, each session of informal learning was carried out within one semester. Students were advised to participate in two weeks of informal learning each month during the period. However, the weekly schedules were subjective to changes dynamically. Table 2 shows the typical weekly schedule of the informal learning community.

## 2.3 Design

The study employed an embedded mixed-method (EMM) design. The rationale for the adoption of EMM research design was to support the findings of one certain type of research

**Table 2. Weekly schedule for informal learning community.**

| Days | Contents | Learning Medium |
|------|----------|-----------------|
| Monday | Weekly Briefing | Forum and Web Application |
| | Workshop & group discussion | Streaming & Messengers |
| Tuesday | Interpreting Show | Streaming |
| | Simulated Conference | Virtual Conference Application |
| Wednesday | Interpreting Technology | Streaming |
| | Workshop and group discussion | Streaming & Messengers |
| Thursday | Students-directed Learning | Decided by students |
| Friday | Weekly Brainstorm | Streaming |
| | Workshop and group discussion | Streaming & Messengers |

method as primary source of data with findings from a supportive and secondary one [36]. Specifically, qualitative data as participants' understandings and quantitative data as numeric presentation of students' involvement in the project were collected as primary and secondary sources of data respectively.

Qualitative approach was adopted in the study to examine students' perceptions of the informal learning community piloted from 2020 to 2022. In the present study, focus group discussion (FGD) with participants was applied to understand participants' perception, attitudes, reflection and experiences in the informal learning amid Covid-19. FGD sessions were carried out at the beginning and end of each semester with 5–8 member each group. In total, eight sessions of FGD were conducted until maturity of the pilot project and saturation of data collection were reached in accordance with the views from expert panel members and the authors.

Supportive and secondary research methods included survey, observation and document analysis. A survey adopted from previous studies measuring students' perception towards an educational project was used to provide quantitative evidence regarding students' level of satisfaction towards the informal learning project [37, 38] (The survey was shown in **S1 Appendix**). Observations of classroom activities and the informal learning project were used to provide a lens from which student's involvement and engagement were examined. Document analysis of student's learning artifact and the grade in summative tests were used to understand the changes of learning outcomes from a longitudinal perspective. All secondary sources of data were collected and analyzed to support the findings of the primary source of data.

### 2.4 Participants

The population of the study was the BTI candidates (N = 455) at XYAFU. Snowball sampling method was adopted in the student participant sampling process for both the qualitative and quantitative inquiries [39]. The reason to choose snowball sampling method is due to the collaborative learning nature of interpreter training. A random sample (n = 18) from the population was first recruited, and their training partners were automatically recruited (n = 36). All participants were Chinese with an average age of 20.5 year. Table 3 demonstrated the demographical information of student participants.

**Table 3. Demographic information of participating students.**

| Gender | Freshmen | Sophomores | Juniors | Seniors | Total |
|--------|----------|------------|---------|---------|-------|
| Female | 6 | 10 | 8 | 4 | 28 |
| Male | 2 | 4 | 2 | 0 | 8 |
| Total | 8 | 14 | 10 | 4 | 36 |

## 2.5 Procedures

All participants were well informed of the design, objectives and data confidentiality measures of the study prior to data collection. Student participants were requested to report their learning progress each month and invited to join a FGD at the end of a training session (which was at the beginning and end of each semester). Each session of for FGD lasted for 45 minutes, with all content audio recorded and transcribed verbatim.

To ensure credibility, two lecturers not directly involved in the project were trained and recruited as moderators for each session of FGD. The discussion followed pre-determined questions in the FGD protocol. Recorded audio and corresponding transcripts of the FGD sessions were member-checked before submitting to the researchers [40].

Additionally, the chief and deputy director of the BTI program were invited to supervise and monitor the data collection processes.

Transcribed and preprocessed data from FGD were analyzed thematically à la procedures from *AMEE Guide No. 131* [41]. Upon the completion of FGD, a codebook was created to guide the exploration for themes from the FGD transcripts (see Table 4 for excerption of the codebook). Two lecturers (W.J. and H.G.) assisted the authors in the coding and thematic analysis process. Disparities in opinions pertaining to the selection and analysis of data were addressed through discussion until consensus was reached. The thematic analysis followed the following six-step procedures: the researcher generated initial codes after getting familiar with the transcribed texts; a reviewing and refinement of the themes were conducted after the completion of searching for themes from coded sources; themes extracted were defined and named before a final report was produced.

At the middle point of each informal learning session, a summative assessment for students' learning outcomes was administered for all participants. A semester-end survey for satisfaction towards the pilot project was conducted. Students were requested to track their learning experiences in the form of learning portfolios for major learning artifacts (e.g., performance in interpreting tasks and interpreters' note-taking in completing tasks) during the informal learning project. All requested data were anonymized by using pseudo names and only retrieved and analyzed for the purpose of research.

The collected and cleaned quantitative data were analyzed with R [42]. Descriptive statistics were used to investigate the trend in changes of different aspects of student's learning over time. R package *ggplot2* was used for data visualization of the quantitative data [43].

**Table 4. Excerpted codebook from the qualitative analysis of FGD.**

| Code | Definition | Purpose of the Code | Sample Quotes |
|---|---|---|---|
| Preparedness | Participants described their readiness to embrace informal learning as a component of regular interpreter training program. | 1. Identify the level of readiness of participants for taking part in the pilot project. 2. Identify the challenges encountered by participants in conducting informal learning. | "[upon the completion of the informal learning project], I felt I am doing good in most activities, so I believe I am prepared to have it in normal learning scenarios. . ." "Digital abilities were the major challenge for me to be fully involved in the online activities" |
| Impact | Participants reflected the influence of online informal learning on their learning. | 1. Identify impact on participants' learning within the activities of the project. 2. Identify the general impact of the project on their learning. 3. Identify the impact of informal learning on formal interpreter training. | "[in activities of the project], I understood the importance of learning from personal experiences from the trainers." "The flexibilities offered by informal learning compensated the rigidity of formal interpreting education, especially in eliciting the students' proactivity." |
| Perceptions | Participants described their understanding and attitudes towards the pilot project of online informal learning. | Identify the perceptions towards the experiences in participating the project. | "[being a student experienced both face-to-face instruction and online training], I am fully convinced that informal learning could be effective in reducing stress in learning interpreting." |

## 2.6 Ethics

This study was approved by the Curriculum Development Committee and Ethics Committee of School of Foreign Languages, Xinyang Agriculture and Forestry University. Written informed consents were obtained from all participants of the study.

# 3. Results

Based on the analysis of FGD with students, the researcher identified three main themes related to the experiences and perception about the informal learning community amid Covid-19: 1). Strength of the pilot project; 2). Roles of Interpreting Trainers; 3). Gains from informal learning. Results of thematic analysis for each theme were described in following subsections.

## 3.1 Strength of the pilot project

For participants of the informal learning community pilot project, consensus was made towards the positive impact of informal learning. Five aspects related to the strength of the project were extracted as subthemes: complementary to classroom learning; trial of alternative learning methods; reduction of stress; digital Literacy; offsetting impact from Covid-19. Table 5 displayed the frequency of subthemes extracted.

**3.1.1 Complementary to classroom learning.** When asked about the relation between informal and formal learning, students reached an agreement in the complementary value of the pilot project. A student commented:

> "*I don't think it will replace our original course, but they co-exist well. I think the informal one is valuable to help the normal course.*" (FGD Session, Nov 2020)

The most appreciated aspects of the strength of the informal learning community in enhancing learning was its wide selection of learning materials, as reflected by a student:

> "*. . . since my participation of the informal learning platform . . . we have access to a large variety of materials of different levels of complexity and difficulties. What's more, we also are now quite familiar with ways and places to find needed materials . . . I can still find what I need in the future.*" (FGD Session, Jun 2021)

Since December 2021, the interpreting trainers began to use a web-based application for the submission, editing, and distribution of pre-processed interpreting materials for participants. The upgrade was well-acclaimed by a participant:

> "*A major change during the implementation of the learning community was the availability of the website. . . Especially when we were away from campus during the Covid. Downloading the materials were just one click away from us. . .*" (FGD Session, Feb 2022)

**Table 5. Subthemes of strength for informal learning.**

| Subthemes | Frequency of themes |
|---|---|
| Complementary to classroom learning | 12 |
| Trial of alternative learning methods | 17 |
| Reduction of stress | 25 |
| Digital Literacy | 26 |
| Offsetting impact from Covid-19 | 14 |

Consensus was reached for the flexibility in the informal learning community. First, students were offered freedom in determining preferred ways of learning. As per the observation, uniform requirements of task completion would not only bore the students but also wear out the enthusiasm of lecturers. Second, the schedules for task completion were elastic. The flexibility in submission or completion of tasks was the core benefit of informal learning. Students generally favored the self-determined model of learning, as a lecturer observed:

"*[the freedom to choose details in learning] was beneficial for students. Instead of causing procrastinating, they were proactive in completing assignments on the contrary. . . .*" *(Conference on the pilot project, Apr 2021)*

Third, there were no compulsory requirement for the participation of learning activities. Students would adjust and plan their participation and learning plans according to their own situation and personal needs. According to a comment in the FGD:

"*I like the design of not forcing us to take part in all learning activities. Some are too hard for me . . . I found that the more freedom I am offered, the more I want to participate.*" *(FGD Session, Oct 2020)*

**3.1.2 Trial of alternative learning methods.** Contrarily to the failure to provide effective learning method in early 2020, the informal learning community offered a platform for experimenting alternative learning methods. The informal learning community advocated the choice of learning methods such as self-directed learning, experiential learning, collaborative learning and assessment for learning. Practicing various relatively less frequently implemented learning models among undergraduates received prominent feedback. A sophomore said:

"*Assessment for learning*?. . . . *I thought the job of assessment were the teachers instead of ours. But after trying it for a few months . . . I have better interpreting since I am familiar with the grading rubrics or standards.*" *(FGD Session, May 2021)*

When asked about the applicability of fusing alternative learning methods in conventional classroom teaching, students expressed their support and interest. A student said:

"*I think we should try some of these new learning methods . . . students were not activated . . . I think we should embrace some of these new ways of learning, even when get back to . . . face-to-face teaching.*" *(FGD Session, Dec 2020)*

**3.1.3 Reduction of stress.** The keywords "burnout" and "stress" were frequently referred to in the FGD with students. In the pandemic, the challenges and pressure in interpreting, even in a simulated environment, were beyond the students' ability to control properly. As a student reported:

"*Stressful, I would say . . . Being a novice learner. I was shivering . . . my brain just goes to blank . . . it was just too hard for me to control myself*". *(FGD Session, Jun 2021)*

Being an agricultural university, the proportion of students coming from rural areas were higher than the average of Chinese universities. Covid-19 brought back to students the burdens to work in farms, caring for infants in family. A female student said:

*"Sometimes I was too tired. I have to look after my newly born sister . . . stay in the shop that my parents run . . . I could hardly focus . . ." (FGD Session, Oct 2020)*

Participants agreed that the "apprenticeship" model in the informal learning helped remedy their stress in interpreting learning as novices. After a brainstorm session, a student said:

*". . . I can handle my stress much better when the lecturer told us his own experience as an interpreter for a conference. I . . . use the tricks he used . . . it worked . . . It feels like learning from a master like in a Kungfu novel . . ." (FGD Session after the brainstorm, May 2021)*

**3.1.4 Digital literacy.**   When asked about the main challenges faced by learning of interpreting during Covid-19, most students agreed that technical limitations and the lack of digital literacy were top factors. Most participants agreed that the use of Information and Communication Technology (ICT) and resources from the Internet facilitated the learning of interpreting by providing the most up-to-date materials and state-of-the-art technologies available for language learning. As a student wrote in her semester summary:

*"One of the most direct gains I received from the learning group is the volume of information. I was amazed in the first day to see how good my lecturer was at collecting useful information . . . He even had a little program online to push new materials directly to our mailbox. I found it cool. I think it is very important for us to master some new technologies." (Student's summary in the learning portfolio, May 2021)*

Perceptions towards the extent to which modern translation/interpreting technologies were used were diverse. Some of the students expressed interest and positive attitudes towards using technology in interpreting learning, while others rejected such opinions:

*Student A: Instant audiovisual interpreting is really awesome. It works like magic. And it is free of charge. . . you can use it for many different materials. I like it, and all I need to do is to read and modify some words or correct some misinterpreting*

*Student B: I know the importance of technology, but I don't think it is wise to use them in learning . . . What is the point for us to learn interpreting . . .? (A group discussion, 2021)*

Though different voices were heard about adopting modern technologies in interpreting learning, majority of the participants agreed that digital literacy was important. In an in-class survey for the core competences for a qualified interpreter, 63.8% (n = 24) of the students chose digital literacy.

**3.1.5 Offsetting impact of Covid-19.**   Most participants in the study shared similar feeling about the difficulties of teaching translation and interpreting in Covid-19. Noticeably, the issue caused by the inadaptability in transforming translation training to online learning were highlighted by most participants among all challenges encountered during the pandemic: *"The first semester was so bad. The interpreting course was just like a listening course"*. However, the role of informal learning was appreciated by most participants:

*Student A: . . . I think the online community would be a solution to solve the problem of pure online teaching. . .*

*Student B: I agree. As the teacher said, we don't have an online training platform for interpreting . . . the learning community was a very steady move. . .*

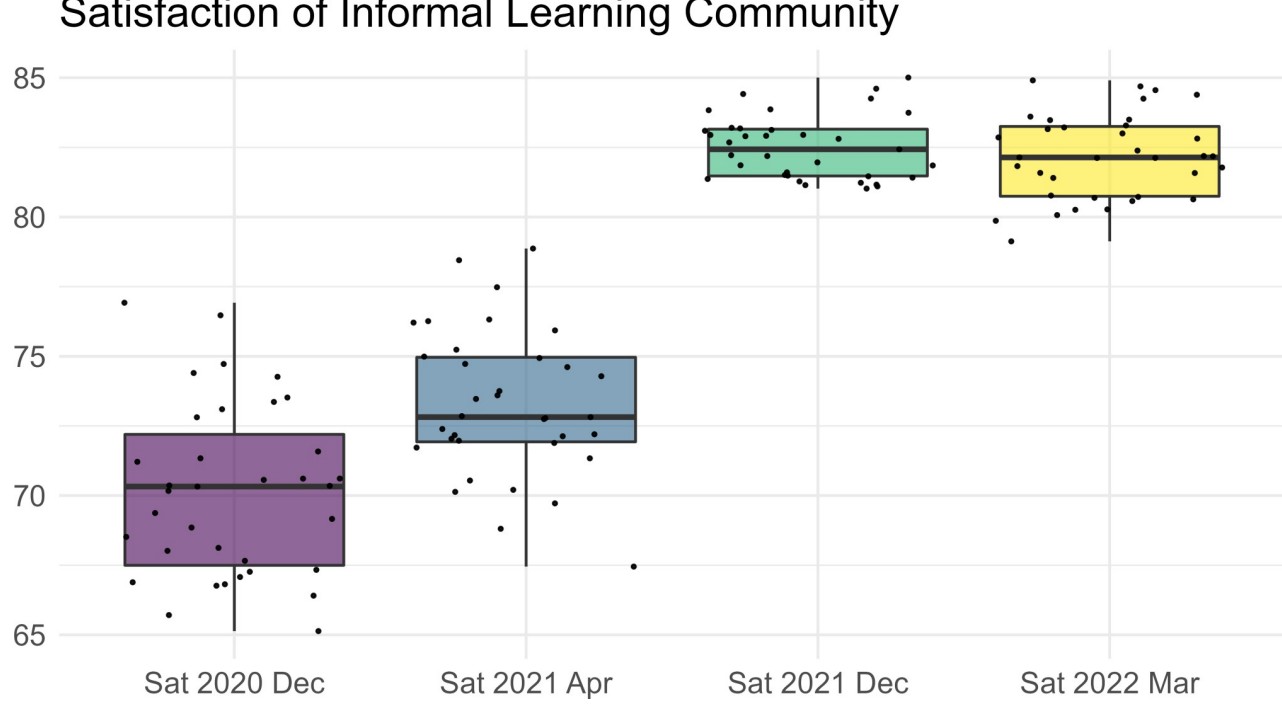

Fig 2. **Results of satisfaction survey after each semester.**

Student A: True. I think we should go on and make it part of our daily learning.

Student C: It would be wonderful if we had this learning model in the beginning of the Covid. (Wechat Group Discussion, Feb 2022)

Students were generally satisfied with the impact of the project. According to the surveys conducted at the end of each semester, the average level of satisfaction grows incrementally (The results of the survey were shown in **S2 Appendix**). See Fig 2 for the satisfaction of the informal learning community during four individual surveys.

### 3.2 Roles of Interpreting Trainers

The pilot project was designed to seek alternative roles of interpreting trainers in the teaching a learning of interpreting. During the one and a half years of pilot project, the involving lecturers co-operated in diverse ways with students. The communication and relationship of trainers and trainees were largely personal. Hence, the roles of interpreting trainers from the perspectives of students vary greatly. Following subthemes (or roles of trainers) were synthesized from the discussion with participating students: mentors, eyeopeners, and assessors. Table 6 displayed the subthemes.

**3.2.1 Mentors.** A frequent mentioned identity of interpreting trainer was "mentor". To most of the participating students, the most valuable strength of the trainer was still their ability of interpreting. As a comment on Weibo, the micro weblog platform, said:

"... I like the lecturer, because he just can do the job right ... Especially the moment he talked about his tricks ..." (Online Group Discussion, Nov 2020)

**Table 6. Subthemes of interpreting trainers' roles.**

| Subthemes | Frequency of themes |
|-----------|---------------------|
| Mentor | 20 |
| Eyeopener | 21 |
| Assessor | 18 |

By referring to "abilities of interpreting", many students referred to professional experiences. A student commented that:

"*The best interpreting teacher will always be those who have done interpreting before. I . . . value the teacher's experience . . . someone without any experience of interpreting . . . had no passion, and his understanding . . . was not perfect . . . not that of a pro". (FGD Session, Jul 2021)*

**3.2.2 Eyeopener.** Another frequently mentioned role or identity of interpreting trainer was "eyeopener". To most participants, an excellent interpreting trainer should be able to broaden the horizon of students. A male student mentioned in a group discussion:

"*The online community really showed me a lot of things that would never be learnt . . . in classroom sessions . . . It was the first time I know an interpreter will not really eat at the dining tables with important figures. . ." (Online Group Discussion, Sep 2021)*

Knowledge beyond the description on textbooks or easily accessed by students were always favored. A comment on the social media account of a trainer wrote:

"*I remember one day, the lecturer showed us how an interpreter prepares his or her note-taking booklets . . . I never know it was just an art . . ." (Social Media Posting, Dec 2020)*

Students reached a consensus on the advantage of informal learning in bringing in new knowledge to interpreting trainees for its instructional flexibility and relatively small audience. According to a female student:

"*I think it might be very different for a teacher to show something to a 50-student class and an 8–10-member online group . . . Student might be puzzled, and he would not know whether he should go on or not . . . For any image we can zoom in and ask about it quickly. For audio and video materials, we can just repeat." (FGD Session, May 2020)*

**3.2.3 Assessor.** The informal learning community for the pilot project was to provide a connection between classroom learning and autonomous learning by providing frequent formative assessment. According to one of the participants:

"*I think it is perfect for us to have the chance to have our own practices by teachers . . . After one year, I have six practices marked by three teachers and many more assessed by myself and my peers. I think I have learnt a lot from this. This might be the best part of the experiment for me." (FGD Session, Nov 2021)*

**Table 7. Subthemes of gains from informal learning.**

| Subthemes | Frequency of themes |
|---|---|
| Learning Efficiency | 11 |
| Interest and Engagement | 20 |
| Language Proficiency | 12 |
| Communicative Abilities | 13 |
| Self-assessment Literacy | 9 |
| Interpreting Competence | 25 |

### 3.3 Gains

In March of 2022, when the pilot project of informal learning community for interpreting trainees entered into the phase of summary and reflection. Students were invited to express their perceptions and understandings of their gains from the pilot project. Surprisingly, gains from students covered not only language and interpreting proficiency but also the building of professionalism and the awareness of importance of self-assessment in learning. Table 7 lists the six subthemes extracted and synthesized.

**3.3.1 Learning efficiency.** The informal learning community for interpreter training were believed to be a booster in learning efficiency:

> . . . accelerator of productivity. I used to be quite slow . . . now, with the help of the informal learning community, we can freely learn by collaborating with each other . . . I can do more things during the same period of time now. (Personal communication, 2022)

Fig 3 shows the reported time spent for each task during a semester. It should be noted that the difficulties of the tasks were based on students' own perception according to uniform difficulty identification standard used in the program.

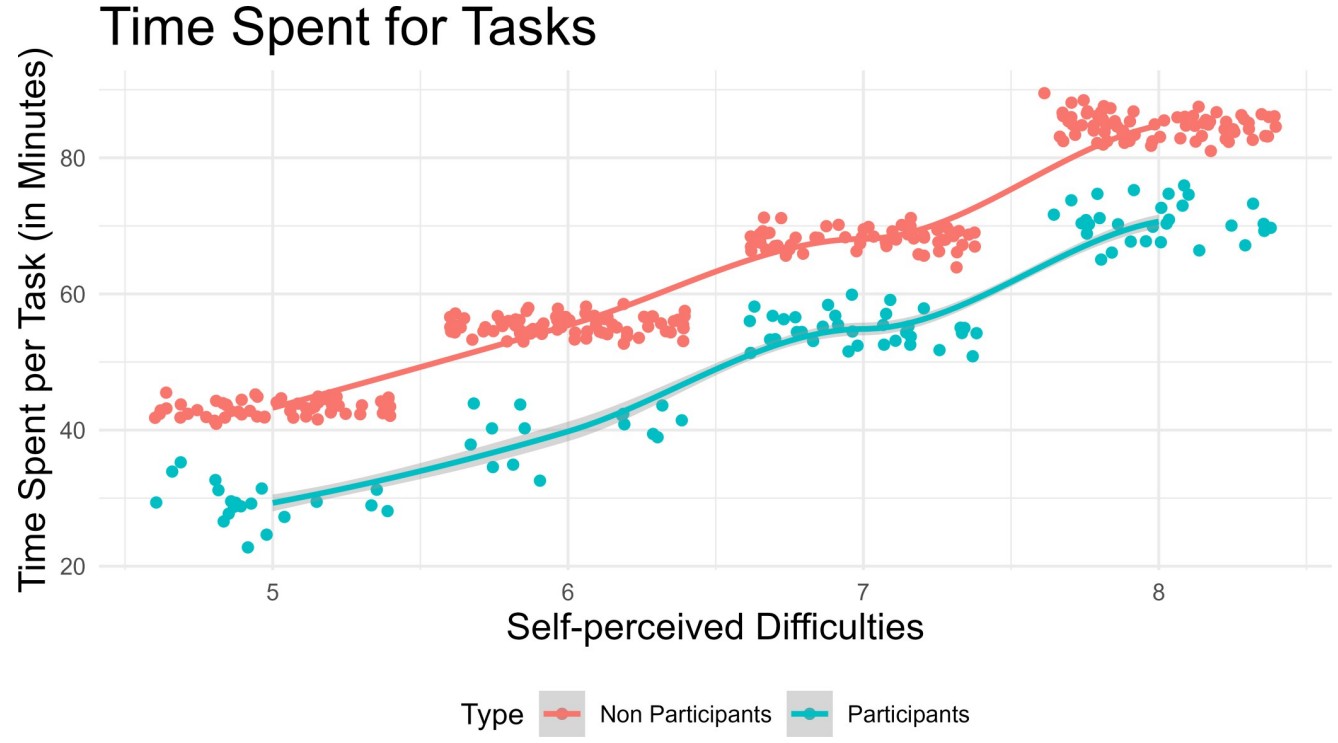

**Fig 3. Time spent for tasks in informal learning.**

**3.3.2 Interest and engagement.**  For most of the participants, the informal learning was effective in raising their interest towards interpreter training. A student commented:

"... *informal learning changed my attitudes. Interpreting included so many aspects of language application. It was actually fun. I am sad for those we were trapped in classrooms. They will not know the fun side of the story.*" (FGD Session, May 2021)

The elatedness of classroom teaching was beyond the original design and assumption towards the impact of informal learning. As commented by the chief director of the BTI program after observing the informal learning community:

"... *The overall involvement for individual student in the learning community was active. But in the classroom teaching, changes were remarkable. Especially for those who were not so active in previous training sessions ... I think the informal learning was positive in classroom learning engagement as well.*" (FGD Session, Sep 2020)

The amplified learning interest and engagement were mutually reinforcing. In commenting on his own performances and mental statuses for interpreting learning with and without the informal learning community, a student said:

"*I noticed my own change in interest ... If I am confident, I don't fear making mistakes. But when I am not interested, I don't have much confidence in myself ... talking about what you have done in the community makes you feel like an experienced one...*" (FGD Session, Nov 2021)

A unison was heard on the possibility of adopting informal learning in classroom among most participants. A representative comment was:

"*I think the way we were asked to learn in classrooms should be changed! ... I have talked with many of my peers, and we are in the same line for this. Interpreting was hard for beginners ... informal ... as well as true 'collaboration' would help greatly.*" (FGD Session, Feb 2022)

**3.3.3 Language proficiency.**  After participation in the informal online learning community, the majority of participants reported a self-perceived progress in language proficiency. As summarized by a male student in online communication with the authors:

"*I think the most apparent progress I have made was about my language abilities ... my progress in language was not limited to English ... I think I have made progress in both English and Chinese...*" (Personal communication, 2022)

**3.3.4 Communicative abilities.**  According to participants, increased chances and frequency of communication between lecturers and students helped to increase their communicative abilities. As reflected by one student:

"*Communication was the most important skill for any language learner ... But informal learning is different. For us, who are the first batch to be recruited, we talk and communicate*

*with each other including the lecturers almost every day. I think I now know how to communicate with peers and instructors, in a highly efficient way. And I am also sure this ability could help my communication with others in other scenarios as well." (FGD Session, Aug 2021)*

**3.3.5 Self-assessment literacy.** Student were informed of the central role of assessment in the informal learning community beforehand. Their experiences during the 1.5 years contributed to an agreement upon the significance of assessment literacy in learning:

"*Assessment was very important in interpreting learning. At the end of the day, what matters is how good you can interpret a given material. But the experience of extensive self-assessment and peer-assessment was important . . . we were using single rubrics and standards. . . it is objective . . ." (Personal communication, 2022)*

In regard to the role of assessment in interpreting learning, students were affirmative. As a group leader said:

"*What matters most is not giving a mark, but why and how . . . This is not only part of the learning process, but also a core part of it. I would say, the experiences in assessment promoted and changed my learning deeply. I know really understand what the word 'assessment' means to us." (FGD Session, May 2021)*

**3.3.6 Interpreting competence.** One of the most frequently mentioned keywords related to the gains from the project in verbatim transcriptions of FGD was "interpreting competence". The BTI program director and the pilot project coordinator commented after observing students' performance of participants and non-participants:

"*The two groups of students were both making progress in using interpreting strategies and skills. But it seems that the groups with experience of the pilot project were generally showing more maturity and skillful handling of the tasks. The differences were even larger, if we examine their interpreting notes. The groups involved in the project were more professional, with more succinct and useful notes taken down." (Conference of the Pilot Project, Jan 2022)*

It was worth noting that progress made in such a short span of time was frequently ignored by students. However, the observation of lecturers was supported by the grades of participants in the latest three iterations of summative assessment for interpreting abilities administered in July 2021, November 2021 and March 2022 respectively. The summative assessment of interpreting competence was designed to test students' performance of interpreting in 8 different aspects. Fig 4 displayed the change of assessment grades of four representative participating students.

## 4. Discussion

The major findings of the present study included: 1) online informal learning could be an effective complement to classroom-based learning; 2) in online informal learning, interpreter trainers' roles multiplied; and 3) the impact of online informal learning on trainees were positive. Put into the broader picture of educational development and pedagogical innovation [28], the findings were noteworthy for improving the quality of interpreter training, and offsetting the negative impact caused by public health emergencies such as Covid-19 [10].

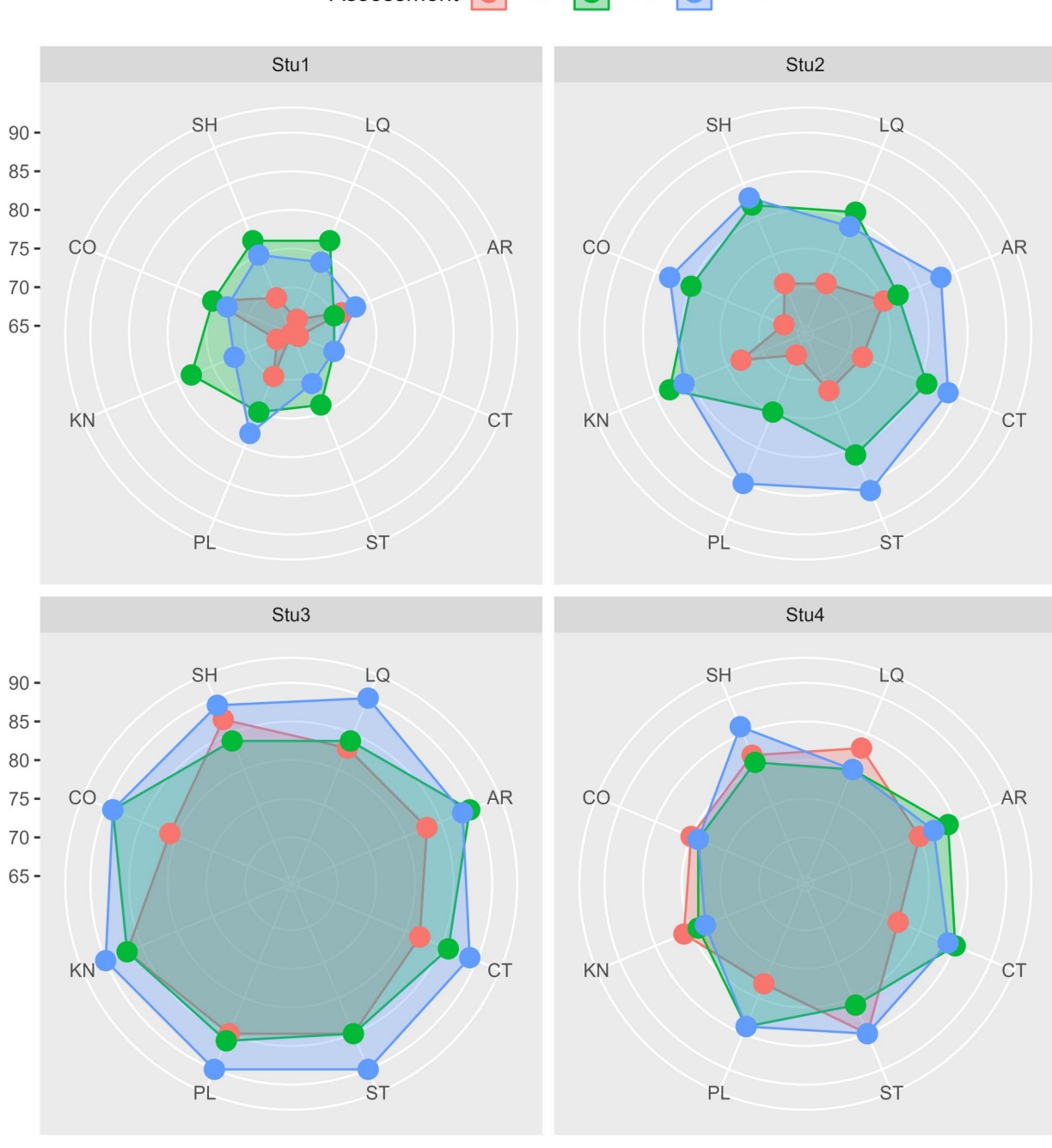

**Fig 4. Grade changes of four students in three assessments.** The interpreting competences assessed included: Stress Handling (SH), Language Quality (LQ), Accuracy (AR), Creativity (CT), Succinctness (ST), Paralinguistics (PL), Knowledge (KN), and Correction (CO).

In the present study, we developed and implemented a pilot online learning project in which students' agency in learning were reinforced. Though situated in an interpreter training program at a Chinese university, the key findings and project outcomes were in tandem with the assertions and claims from previous literature regarding the impact and effectiveness of adopting informal learning in higher education. Trinidad argued that among major issues

encountered in school organization activities, the mental health of students was a priority for decision-makers in the pandemic [23]. In our study, we identified that classroom-based formal instruction was relevantly less efficient in reducing mental stress of interpreting learners. However, with the implementation of the informal learning project, students reported a positive impact of the informal learning environment in curbing their tension and pressure. Similarly, Segers and his colleagues consolidated the advantage of informal learning in adopting to new trends of learning requirements such as knowledge density [26]. In our study, the value of informal learning in supporting formal learning was verified. According to the responses of the participants, the combination of both formal learning and informal learning was favored for offering the opportunities to learn beyond the limited time prescribed by the interpreter training program. Most importantly, the implementation of informal learning in the present study offered trainers and trainees the chance to test the effectiveness of alternative learning strategies. The experiences from participants were in line with the claims from relevant studies, students were empowered in the learning process for the exposure to student-centered learning environment [28], relaxed learning atmosphere [32] and the convenience brought by a technology-enhanced learning approaches [31].

In regard to the impact of online informal learning on the interpreter training program, the study revealed that multiple roles of trainers were expected, and the pilot project was generally believed to be able to enhance students' core competence as future interpreters. In accordance with findings from the research conducted by Mehrvaz and his colleagues, informal learning implemented online could be positive for improving students' digital literacy and academic performance [33]. The endeavor to employ blended learning by resorting to online learning resources during the study was favored by participants in a similar fashion to the research of Xu et al. [21]. In the present study, participants reported gains in both core competencies for interpreters (e.g., language abilities and interpreting-specific skills) and digital literacy. Additionally, student self-perceived their increase in learning motivation and interest for the interpreter courses which were believed to be challenging for learners during the pandemic [16]. Most importantly, we have identified the fact that multiple identify of trainers during interpreter training were expected by participants. This was in line with the nature of interpreter training which originally was conducted in an "apprenticeship-based model" in which students followed the experiences of trainers as role models [44]. The finding unveiled the importance of trainee-trainer relationship, identification of the roles of interpreter trainers during education and the teacher learning of interpreter trainers for abilities in dealing with the out-of-class needs of trainees (e.g., reduction of stress, exchange with trainers about professional experiences, etc.). However, these topics were not sufficiently discussed by scholars interested in interpreting studies currently. Being an evaluation of a pilot project, our study would be significant in expanding the scope and foci of relevant fields.

Additionally, unexpected findings were identified during the inquiries about the perceptions regarding the possibility to implement informal learning officially in the interpreter training program. Some students were unsure about the sustainability of the informal community in action among a larger population of participating students. The lack of confidence went against the findings from existing documented cases of implementing informal learning as a complementary approach to support the formal learning [29, 30, 32]. However, the issue could be attributed to student's lack of experience in alternative learning methods and the inadequacy of teacher education for improvement of trainers' pedagogical literacy. Additionally, it should be noted that the design and implementation of informal learning was part of a comprehensive restructuring of the interpreting program at XYAFU. Consequently, relevant adjustments in curriculum, teaching methods and recalibration of training objectives were undergoing [37]. As suggested by Temban and his colleagues, pre-determined and prescriptive

learning objectives and curricular settings used in formal learning environment hindered the development and augmentation of informal learning [45]. The doubts from participants should be given attention in following-up research and investigations.

## 4.1 Implications and recommendations

Since the impact of Covid-19 on tertiary education continued, attention should be paid to the specific issues and opportunities encountered in interpreter training: readiness of novice and pre-service trainers, psychological health of trainers and trainees, the reevaluation of lecturers' role in training and the adopting of innovative learning methods in interpreter training.

First, tailored teacher education should be implemented for trainers, especially novice or pre-service trainers. In the pilot project, the most frequently mentioned expectations from students were digital literacy and interpreting professionalism. The lack of lecturers with industrial experience or instructors with high level of digital literacy were a common issue faced by BTI programs across China [46]. To solve the problem, qualified lecturers should be recruited, while in-service trainers should be retrained [47]. However, the in-demand teacher education for interpreting trainers were lagging in China, where new lecturers received general-purpose pre-service trainings with no consideration of the uniqueness and characteristics of his/her career. Hence, the authors would recommend developing and implementing tailored teacher learning program for interpreter trainers within a program. Frequent and well-organized teacher learning activities (e.g., collegial learning [48], collaborative curriculum development [49]) would be vital for the fostering of competent trainers.

Second, the mental wellbeing of trainers and trainees should be catered for. During the pandemic, the psychological heath of both trainers and learners were at risk [50]. The amalgamation of family burden, fear of infection, and academic pressure, and the innate intensity of interpreter training contributed to the excessive burnout among trainees. However, the availability of stress reduction or educational care within a BTI program was hard to be found in China. The present study confirmed the value of informal learning in reducing stress among interpreting learners. Consequently, the authors would recommend embedding regular stress control or reduction instructions in a training program. For example, the success in adopting mindfulness training to calm down the trainees from previous studies could be followed [51, 52]. Additionally, frequent and continuous tracking of trainees' mental status would be imperative.

Third, the trainer-trainee relationship could be redefined. In conventional settings, the role trainers played in interpreter training were similar to lecturers in other language courses [53]. Nonetheless, in the study, the trainer-trainee relation was released from conventional norms and reconstructed along the practices of informal learning. In an informal learning environment, the professionalism and competences of the trainer were maximized. Extra abilities such as digital literacy or personal charisma were highlighted. The expanded roles of interpreting trainers aroused students' interest to follow. For undergraduate trainees, multiform identities of trainers and diversified learning approaches would benefit the students. Students with ambitions to pursue the career of interpreter after graduation could be further stimulated by the iconic role of trainers. The authors would recommend frequent trainer-trainee extra-curricular activities and periodical need analysis of trainees in the program.

Finally, innovation in learning methods should be encouraged. To improve the quality of BTI education, headways should be made in implementing and adapting alternative learning approaches. The pilot project revealed that students were generally more satisfied with learning methods that stimulated their agency and engagement. Globally speaking, efforts have been made in recent years by employing alternative learning approaches in interpreter

training, from computer-assisted learning, assessment-driven learning to student-centered learning [54–57]. Nevertheless, Chinese BTI programs lagged behind the innovation trend in teaching and learning. In most of the undergraduate training settings in China, lecturer-centered training remained the mainstream. The authors would recommend program decision-makers and interpreter trainers to incorporate alternative learning methods and conventional training in a systematic manner. The experiences learnt from the present study would be beneficial in training model construction, development and implementation of innovative learning methods, and evaluation of the updated training schemes.

## 4.2 Limitations and future perspectives

As an effort to pilot the impact of informal learning in interpreter training and an attempt to support teaching and learning amid Covid-19, the study faced one major limitations of limited number of participants. The number of students and lecturers recruited for the pilot project was intentionally controlled. The relatively smaller size of sample restricted the reliability of the pilot project. The issue was remedied by adopting a longitudinal design, in which several rounds of data collection and analysis occurred. The online informal learning community adopted an "on-the-fly" modification mechanism to adjust the learning environment to the authentic needs for learning. Additionally, an expert panel was also invited to monitor and guide the project.

Looking for the future, the project left much space for expansion and continuation of research. In the Chinese translation/interpreter training context, the teaching and learning of translation and interpreting at undergraduate level were indispensable. The positive impact of informal learning in promoting students' learning outcome and agency were possible to be transferred to those of translation. For example, lecturers have tried to shift the teaching of computer-assisted translation (CAT) and machine translation (MT) to student-centered or project-based learning models in recent years [58]. For these courses, informal learning could be adopted in the form of hackathon or amateur projects. On the other hand, there were many different aspects related to interpreter training that were not discussed in the pilot project. For example, status and factors of burnout of leaners and lecturers amid Covid-19 was not examined in detail.

## 5. Conclusion

The continuing impact of Covid-19 on interpreter training put undergraduate interpreter training programs at stake. Lack of readiness and shortage in alternative teaching and learning approaches caused the rather disillusioned educational outcomes in the initial attempt to only use live streaming as the means of remote education. The present study was an innovative effort in development online informal learning community to address existing issues of interpreter training amid Covid-19. The perception and satisfaction towards the informal learning community were positive and encouraging. The study also contributed to the reevaluation of trainer's roles in interpreter training. Retrospectively, many underlying issues hindering the healthy development of interpreter training was caused by conservativeness in trying and developing innovative training and learning approaches. The results propelled trainers, program decision makers, curriculum designers and university administrators to rethink education during the pandemic, and how to seize the opportunity to reconstruct and innovate the existing curriculum and pedagogical patterns. The study was limited by the number of participants. Nevertheless, the contributions the theoretical and practical aspects of our understanding in enhancing learning engagement and academic achievement by incorporating

alternative learning approaches would further expand our cognition of the new horizon of education towards post-pandemic recovery.

## Supporting information

**S1 Appendix. Satisfaction survey for the pilot project.**
(DOCX)

**S2 Appendix. Results of the satisfaction survey.**
(DOCX)

## Acknowledgments

Warmest gratitude goes to Ms. Chenjin Jia for her enlightenment and encouragement in a conversation that shaped the backbones of the study, to all participating students and lecturers for their enthusiasm and persistence in the pilot project, to Mrs. Junyue Wang and Ms. Hualing Gong for being the coders of the qualitative data analysis. We would also like to express our thanks to the editor and both reviewers for their constructive feedback.

## Author Contributions

**Conceptualization:** Da Yan.

**Formal analysis:** Da Yan.

**Investigation:** Da Yan.

**Methodology:** Da Yan.

**Project administration:** Da Yan.

**Supervision:** Qiongqiong Fan.

**Validation:** Da Yan.

**Visualization:** Da Yan.

**Writing – original draft:** Da Yan.

**Writing – review & editing:** Qiongqiong Fan.

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
