## [Decision Letter · Decision Letter 0]

21 Sep 2022

PONE-D-22-21532Online informal learning community for interpreter training amid COVID-19: a pilot evaluationPLOS ONE

Dear Dr. Yan,

Thank you for submitting your manuscript to PLOS ONE. After careful consideration, we feel that it has merit but does not fully meet PLOS ONE’s publication criteria as it currently stands. Therefore, we invite you to submit a revised version of the manuscript that addresses the points raised during the review process.

We look forward to receiving your revised manuscript.

Kind regards,

Omar Mohammad Ali Khraisat, Associate Professor

Academic Editor

PLOS ONE

Journal Requirements:

2. Peer review at PLOS ONE is not double-blinded (https://journals.plos.org/plosone/s/editorial-and-peer-review-process). For this reason, authors should include in the revised manuscript all the information removed for blind review.

3. Thank you for including your ethics statement:  "N/A".   

For studies reporting research involving human participants, PLOS ONE requires authors to confirm that this specific study was reviewed and approved by an institutional review board (ethics committee) before the study began. Please provide the specific name of the ethics committee/IRB that approved your study, or explain why you did not seek approval in this case.

4. Please provide additional details regarding participant consent. In the ethics statement in the Methods and online submission information, please ensure that you have specified (1) whether consent was informed and (2) what type you obtained (for instance, written or verbal, and if verbal, how it was documented and witnessed). If your study included minors, state whether you obtained consent from parents or guardians. If the need for consent was waived by the ethics committee, please include this information.

6. Thank you for stating the following financial disclosure: 

"The authors received no specific funding for this work."

7. We note that you have stated that you will provide repository information for your data at acceptance. Should your manuscript be accepted for publication, we will hold it until you provide the relevant accession numbers or DOIs necessary to access your data. If you wish to make changes to your Data Availability statement, please describe these changes in your cover letter and we will update your Data Availability statement to reflect the information you provide.

8. Please upload a new copy of Figure 4 as the detail is not clear. Please follow the link for more information:

https://blogs.plos.org/plos/2019/06/looking-good-tips-for-creating-your-plos-figures-graphics/

https://blogs.plos.org/plos/2019/06/looking-good-tips-for-creating-your-plos-figures-graphics/

Reviewers' comments:

Reviewer's Responses to Questions

**Comments to the Author**

1. Is the manuscript technically sound, and do the data support the conclusions?

Reviewer #1: Yes

Reviewer #2: Yes

2. Has the statistical analysis been performed appropriately and rigorously? 

Reviewer #1: Yes

Reviewer #2: Yes

3. Have the authors made all data underlying the findings in their manuscript fully available?

Reviewer #1: Yes

Reviewer #2: Yes

4. Is the manuscript presented in an intelligible fashion and written in standard English?

Reviewer #1: Yes

Reviewer #2: Yes

5. Review Comments to the Author

Reviewer #1: Abstract

Line 11: Term “reviewed” is the author referring to assessment or the term “viewed” referring to perception. The subsequent statement seemingly refers to perception vs assessment and the most appropriate term may be “viewed”

Line 18,135, 312: “N” usually refers to a population size while “n” refers to sample size. Thirty-six is in reference to sample size and thus lower case “n” should be used.

Introduction

Line 33: Lack of coherence between the succeeding statement. The new emerging issues rather than the amplification of underlying issues should follow the first statement.

Line 78, 103, 468 etc: from the programmatic perspective, I would use impact of the .. rather than the effect.

Methods

Line 111, 133; I feel it the rest of the manuscript does not reflect the measurement of “effectiveness” would substitute it with “impact”. Or you may want to provide operational definition of “effectiveness” Additionally, use of “experimental informal…” I find it misleading. When I read that, as a reader, I am looking for rigorous quantitative methodology with comparative groups which is not the case. I would leave it to the “pilot” project as used in some instances. Line 113 and 114, are more appropriate.

Line 171: Review “… qualitative data as numeric….”, this should be quantitative

Line 173-177: Abbreviate Focus group discussion (FGD) and use acronym subsequently. Did you have any gender considerations in the FGDs? How many FGDs were conducted in total and what was the stopping rule?

Line 184-191: It is not clear whether snow balling sampling was done both qualitative and quantitative inquiries.

Line 192-194: The content in this paragraph seems to miss a heading as the content does not match well with the tittle participant on line 183.

Line 198: This takes account of FGDs conducted at the end of the semester and yet earlier on line 176 and 177, this took account of FGDs at the beginning and end of the semester

Line 204: It is not clear of what media participants checked. Did they check the recording or the transcript or both?

Line 206-210: Who was involved in this process and if they are investigators, did you examine your own role, potential bias and influence during analysis and selection of data for presentation? It is also good practice to provide a code book as a supplementary material

Line 211-215: What were the measurements for satisfaction and what are the major learning artifacts that were tracked.

Results

General: Most of the quotes don’t have source like cited on line 309

Line 222, 233: Mention the pilot project

Line 248: Consensus vs consent,

Line 249-250: Observation was not mentioned in the methodology; the interpretation seems out of context as the cited quote does not illustrate that

Line 260-261: italicize

Line 265: First time to come across project-based learning. Earlier in the methods, you mentioned about three pillars of which this was not one of them, even the conceptual diagram does not reflect this. Clarify.

Line 282: Being an agricultural university--- is a potential identifier. I take note you would like to maintain anonymity. I take note that the author is from an agricultural university, whose name is mentioned and thus can link to the potential university where data was collected. On a second thought, this is good practice that other institutions of learning should learn from, I am curious why we are making the university has to be anonymous.

Line 312: would recommend this format 24/36 (63.8%)

Line 314: the tittle of the subtheme should be consistent with the one in table 4

Line 333: project vs experiment

Line 338: Table 5 should be consistent with line 337 interms of number of sub themes; 3 vs 4. Terms rater vs assessor

Line 375-379: does this fit well with the sub theme?

Discussion on wards

Hard to read sections. Reader un friendly

Discussion

General comment: the discussion is insufficient. Any comparative evaluation with other studies

Line 467: Though vs thought

Implications

General comment: There are sections that best fit for result sections. Some of the statements are not consistent with methods and results.

Recommendations

General comment: Some of the recommendations are not consistent with the results.

Reviewer #2: Here are some small edits that the writers should make to the paper that has been reviewed and attached. However, the work's quality is good, and after a short edit, it can be accepted for publishing, barring a few minor corrections. This is a well-written piece, and I applaud the authors for their efforts. The introduction and method sections need some small tweaks, but all other sections are well-organized and explained.

• extremely well-structured writing. It would be fantastic if the authors briefly described informal learning in section 1.1 before beginning to describe it in detail in section 1.2.

• Methodology

The authors did a good job of explaining their research methodology, but in section 2.3, lines 170, 171, and 172, I would advise double-checking the sources of the qualitative and quantitative data because the authors only mentioned the qualitative, and I think line 171 will actually be presenting quantitative data as a numeric presentation.

6. PLOS authors have the option to publish the peer review history of their article (what does this mean?). If published, this will include your full peer review and any attached files.

Reviewer #1: No

Reviewer #2: **Yes: **Nahida Akter

---

## [Author Response · Author response to Decision Letter 0]

26 Sep 2022

Dear Assoc. Prof. Omar Khraisat,

Greetings, 

Thanks for being the editor of the manuscript under the consideration of PLOS ONE entitled Online Informal Learning Community for Interpreter Training Amid COVID-19: A Pilot Evaluation (PONE-D-22-21532). After receiving the reviewers’ comments, the authors began a thorough amendments and revision of the original submission. 

We would like to extend the warmest thanks to both of the reviewers for their valuable comments and constructive suggestions. The manuscript has been revised in accordance with these comments. Changes made in response to the reviewer are described as follows.

Response to the Comments from Reviewer #1:

First, in regard to the comments and suggestion which only needs a relatively small proportion of contents to be revised, our responses are as follows:

Line 11: Term “reviewed” is the author referring to assessment or the term “viewed” referring to perception. The subsequent statement seemingly refers to perception vs assessment and the most appropriate term may be “viewed”

Response: Thanks for the suggestion. The word “viewed” was used in the revised manuscript. 

Line 18,135, 312: “N” usually refers to a population size while “n” refers to sample size. Thirty-six is in reference to sample size and thus lower case “n” should be used.

Response: Thanks for pointing out the issue. The distinction between “N” and “n” was made in the revised manuscript. 

Line 33: Lack of coherence between the succeeding statement. The new emerging issues rather than the amplification of underlying issues should follow the first statement.

Response: Thanks for pointing out the issue. The sentence order was rearranged to avoid the ambiguity in the original manuscript. Please see LINE 33-36 for the amendments. 

Line 78, 103, 468 etc: from the programmatic perspective, I would use impact of the .. rather than the effect.

Response: Thanks for the suggestion. Impact was used to replace effect in the manuscript in all similar occasions. 

Line 111, 133; I feel it the rest of the manuscript does not reflect the measurement of “effectiveness” would substitute it with “impact”. Or you may want to provide operational definition of “effectiveness” Additionally, use of “experimental informal…” I find it misleading. When I read that, as a reader, I am looking for rigorous quantitative methodology with comparative groups which is not the case. I would leave it to the “pilot” project as used in some instances. Line 113 and 114, are more appropriate.

Response: Thanks. We agree that for qualitative-dominant research as the paper, the word “effectiveness” was misleading to the readers. “Impact” was used to replace effectiveness in the manuscript in all similar occasions. 

Line 171: Review “… qualitative data as numeric….”, this should be quantitative

Response: Thanks for pointing out the typo. The correction was made accordingly. 

Line 173-177: Abbreviate Focus group discussion (FGD) and use acronym subsequently. Did you have any gender considerations in the FGDs? How many FGDs were conducted in total and what was the stopping rule?

Response: Thanks for suggesting the abbreviation. We have adopted your suggestion in the manuscript. Gender issues were not considered in the FGD, however, as in many language education settings, the population was female dominant. Regarding the rounds of FGD conducted in the research, please kindly refer to LINE 178-180 for the description about the total number of FGD sessions conducted and the stopping rule for the FGD. An excerption was as follows: 

In total, eight sessions of FGD were conducted until maturity of the pilot project and saturation of data collection were reached in accordance with the views from expert panel members and the authors. 

Line 184-191: It is not clear whether snow balling sampling was done both qualitative and quantitative inquiries.

Response: Thanks for pointing out. The sampling method was used for both strands of research. As the participants were the student involved in the informal learning project. All corresponding data collection and analysis were conducted with them as the participants. 

Line 192-194: The content in this paragraph seems to miss a heading as the content does not match well with the tittle participant on line 183.

Response: Thanks for the suggestion. In the revised manuscript, the paragraph now falls into a separate section entitled “Ethics”. Please refer to section 2.6 for details. 

Line 198: This takes account of FGDs conducted at the end of the semester and yet earlier on line 176 and 177, this took account of FGDs at the beginning and end of the semester

Response: Thanks for the pointing out the issue. The FGD were conducted at both the beginning and end of the semester. In the updated manuscript, all relevant information were checked and revised for internal consistency.

Line 204: It is not clear of what media participants checked. Did they check the recording or the transcript or both?

Response: The media checked by participants involved both the audio recording and the transcript verbatim.

Line 206-210: Who was involved in this process and if they are investigators, did you examine your own role, potential bias and influence during analysis and selection of data for presentation? It is also good practice to provide a code book as a supplementary material

Response: Thanks for the pointing it out. We agree that this is very important for the qualitative research within the paper. Please check LINE 212-214 for the amended information regarding the issue. 

As to the codebook, a sample codebook was added to the manuscript as TABLE 4. The table and relevant descriptions could be found in the manuscript in LINE 211-212. 

Line 211-215: What were the measurements for satisfaction and what are the major learning artifacts that were tracked.

Response: Thanks. We agree that the clarification was important for the manuscript. The measurement of satisfaction was with a survey adopted from the instrument used in previous studies. As to the contents of learning artifacts to be studied, the most important was students’ performance in interpreting tasks and their notetaking (which was a very important learning artifact in an interpreter training setting). The details could be found in the revised manuscript at LINE 181-184 and LINE 224-225.

General: Most of the quotes don’t have source like cited on line 309

Response: Thanks for the pointing it out. We agree that inconsistency in citing the quotes was causing trouble. In the updated manuscript, all sources were amended. Please refer to the quotes.

Line 222, 233: Mention the pilot project

Response: Thanks for the pointing it out. However, we were not very clear about the suggestion of “mentioning the pilot project”. In the revision, we changed the title of section 3.1 from “Strength” into “Strength of the Pilot Project” for clarity. If the solving does not solve the issue you have suggested, please let us know. 

Line 248: Consensus vs consent,

Response: Thanks for the pointing out the typo. It has been corrected, and we are sorry for the mistake. 

Line 249-250: Observation was not mentioned in the methodology; the interpretation seems out of context as the cited quote does not illustrate that

Response: Thanks for the comments. We have amended “observation” as a supportive research method in the methodology. Please refer to LINE 184-185 of the updated manuscript for details. 

In regard to the out of context quote from the participating student, the quote is now substituted by a quote from a lecturer involved in the project based on her observations of student’s involvement and activities within the project. Please refer to LINE 268-271 for details. 

Line 260-261: italicize

Response: Thanks for the pointing out the issue. It has been reformatted accordingly. 

Line 265: First time to come across project-based learning. Earlier in the methods, you mentioned about three pillars of which this was not one of them, even the conceptual diagram does not reflect this. Clarify.

Response: Thanks for the pointing out the issue. We agree that is a very important one and is causing inconsistency between different sections of the paper. 

We have replaced “project-based learning” with “experiential learning” which is more appropriate for the study, especially in line with the conceptual model of the pilot project. However, we would like to point out that in the original design, project-based learning was actually the main approach to implement “experiential learning”, given the nature of interpreting education and common practice within the discipline. 

In a nutshell, we are indebted to the reviewer for pointing out the issue as we were apparently taking it for granted in the original composition. 

Line 282: Being an agricultural university--- is a potential identifier. I take note you would like to maintain anonymity. I take note that the author is from an agricultural university, whose name is mentioned and thus can link to the potential university where data was collected. On a second thought, this is good practice that other institutions of learning should learn from, I am curious why we are making the university has to be anonymous.

Response: Thanks for the pointing out the issue. We were not consistent in choosing to make the affiliations anonymized or exposed. After a series of repeated discussion between the authors, we would like to adopt the suggestions from the reviewer and make the information public. 

Line 312: would recommend this format 24/36 (63.8%)

Response: Thanks for the suggestions, now the number were presented as “63.8% (n=24) of the students chose digital literacy”. Please kindly refer to LINE 329 for detail. 

Line 314: the tittle of the subtheme should be consistent with the one in table 4

Response: Thanks for the pointing it out. And it is corrected as suggested. 

Line 333: project vs experiment

Response: Thanks for the pointing out the typo. It has been corrected, and we are sorry for the mistake. 

Line 338: Table 5 should be consistent with line 337 interms of number of sub themes; 3 vs 4. Terms rater vs assessor

Response: Thanks for the pointing it out. And it is corrected as suggested. As to the choice between rater and assessor, we choose to use “assessor” as the role played by trainers is not limited to rating student’s performance. 

Line 375-379: does this fit well with the sub theme?

Response: Thanks for the pointing it out. The quote was removed from this section. And in the discussion, the doubts from students were treated as “unexpected findings” of the research. 

Discussion on wards: Hard to read sections. Reader unfriendly

Response: Thanks for the comments. After systematic revision of the discussion section. We would attribute the “reader unfriendliness” to over-extended discussion related to decision-making and educational policies in interpreter training. In the revised manuscript, all discussion were based on the findings presented in the previous section. And all off-topic and out-of-context discussion were removed. Please kindly refer to and review the overhauled discussion section for the details. 

Discussion: General comment: the discussion is insufficient. Any comparative evaluation with other studies

Response: Thanks for the comments. We agree that the discussion in the original manuscript was not enough. In the updated version, the part was totally rewritten. Now a comparative stance against existing literature was held by the authors. Please kindly refer to the revision for details. 

Line 467: Though vs thought

Response: Thanks for the pointing out the typo. It has been corrected, and we are sorry for the mistake. 

Implications: General comment: There are sections that best fit for result sections. Some of the statements are not consistent with methods and results.

Response: Thanks for the comments. We agree that some part in the implication and recommendations were not strictly based on the finding of the paper. Consequently, they were not tightly integrated with the context of the discussion. In the revision, we merged the implication and the recommendations. All claims are now based on the findings of the research, and explicit recommendations and implication were preferred. 

Recommendations: General comment: Some of the recommendations are not consistent with the results.

Response: Thanks for the comments. As the recommendations and the implications were merged in the revision, please kindly refer to the response in the previous comments. 

Response to the Comments from Reviewer #2:

Comment 1: It would be fantastic if the authors briefly described informal learning in section 1.1 before beginning to describe it in detail in section 1.2.

Response: Thanks for the comments. The suggestion was valuable, and we are happy to revise accordingly. In the revision, we added a background description of the current status of using informal learning online. Please kindly refer to LINE 41-43 for details. 

Comment 2: The authors did a good job of explaining their research methodology, but in section 2.3, lines 170, 171, and 172, I would advise double-checking the sources of the qualitative and quantitative data because the authors only mentioned the qualitative, and I think line 171 will actually be presenting quantitative data as a numeric presentation.

Response: Thanks for the suggestion and complement. We have noticed the mistake and corrected accordingly.

---

## [Decision Letter · Decision Letter 1]

24 Oct 2022

Online informal learning community for interpreter training amid COVID-19: a pilot evaluation

PONE-D-22-21532R1

Dear Dr.,

We’re pleased to inform you that your manuscript has been judged scientifically suitable for publication and will be formally accepted for publication once it meets all outstanding technical requirements.

Kind regards,

Omar Mohammad Ali Khraisat, Associate Professor

Academic Editor

PLOS ONE

Additional Editor Comments (optional):

Reviewers' comments:

Reviewer's Responses to Questions

**Comments to the Author**

1. If the authors have adequately addressed your comments raised in a previous round of review and you feel that this manuscript is now acceptable for publication, you may indicate that here to bypass the “Comments to the Author” section, enter your conflict of interest statement in the “Confidential to Editor” section, and submit your "Accept" recommendation.

Reviewer #1: (No Response)

Reviewer #2: All comments have been addressed

2. Is the manuscript technically sound, and do the data support the conclusions?

Reviewer #1: Yes

Reviewer #2: Yes

3. Has the statistical analysis been performed appropriately and rigorously? 

Reviewer #1: Yes

Reviewer #2: Yes

4. Have the authors made all data underlying the findings in their manuscript fully available?

Reviewer #1: Yes

Reviewer #2: Yes

5. Is the manuscript presented in an intelligible fashion and written in standard English?

Reviewer #1: Yes

Reviewer #2: Yes

6. Review Comments to the Author

Reviewer #1: General comments;

Thanks for addressing the previously cited comments. they were all well addressed. However, i have comments arising from citation of the quotes in the results section.

It would be helpful to add the FGD number, sex and whether they were freshmen, sophores, juniors or seniors. this will help thereader appreciate representation of the quotes across all participants. Eg. M-FGD 008, Freshman, May 2021; to mean that this quote is from a male from FGD number 8, a fresh man ans was interviewed in may 2021.

Other wise i an not certain if this data cuts across the data collected or could be limited to one or two participants.

Line 327 you mention a group discussion, subsequently, you qualify the group discussion to online, was the group discussion in reference on line 327 an online discussion or otherwise?

Line 327, 406,441,457 You take note of the source of the quote and the year without mention of the month which is not consistent with other format of the "source", "month" and "year" . Could number the group discussions as well and perharps adopt the format referenced about

Line 323-325, 326-327, 337-343; You pull the quotes and cite the source at ago. I would recommend citing each quote separately using the recommended formart.

Reviewer #2: They addreesed all the issues I pointed out before, so it can be recommended to accept thier work and Best wishes for the authors.

7. PLOS authors have the option to publish the peer review history of their article (what does this mean?). If published, this will include your full peer review and any attached files.

Reviewer #1: No

Reviewer #2: **Yes: **Nahida Akter

---

## [Editor Report · Acceptance letter]

26 Oct 2022

PONE-D-22-21532R1 

Online informal learning community for interpreter training amid COVID-19: a pilot evaluation 

Dear Dr. Yan:

I'm pleased to inform you that your manuscript has been deemed suitable for publication in PLOS ONE. Congratulations! Your manuscript is now with our production department. 

Kind regards, 

on behalf of

Dr. Omar Mohammad Ali Khraisat 

Academic Editor

PLOS ONE